# Synchronization of a Passive Oscillator and a Liquid Crystal Elastomer Self-Oscillator Powered by Steady Illumination

**DOI:** 10.3390/polym14153058

**Published:** 2022-07-28

**Authors:** Kai Li, Fenghui Gan, Changshen Du, Guojun Cai, Junxiu Liu

**Affiliations:** 1Anhui Province Key Laboratory of Building Structure and Underground Engineering, Anhui Jianzhu University, Hefei 230601, China; kli@ahjzu.edu.cn; 2College of Civil Engineering, Anhui Jianzhu University, Hefei 230601, China; ganfenghui2022@163.com (F.G.); changshendu@yeah.net (C.D.); focuscai@163.com (G.C.)

**Keywords:** synchronization, self-oscillator, liquid crystal elastomer, passive oscillator, optically-responsive

## Abstract

Self-oscillators have the advantages of actively harvesting energy from external steady environment, autonomy, and portability, and can be adopted as an engine to drive additional working equipment. The synchronous behavior of self-oscillators and passive oscillators may have an important impact on their functions. In this paper, we construct a self-oscillating system composed of a passive oscillator and an active liquid crystal elastomer self-oscillator powered by steady illumination, and theoretically investigate the synchronization of two coupled oscillators. There exist three synchronous regimes of the two coupled oscillators: static, in-phase, and anti-phase. The mechanisms of self-oscillations in in-phase and anti-phase synchronous regimes are elucidated in detail by calculating several key physical parameters. In addition, the effects of spring constant, initial velocity, contraction coefficient, light intensity, and damping coefficient on the self-oscillations of two coupled oscillators are further investigated, and the critical conditions for triggering self-oscillations are obtained. Numerical calculations show that the synchronous regime of self-oscillations is mainly determined by the spring constant, and the amplitudes of self-oscillations of two oscillators increase with increasing contraction coefficient, light intensity, and spring constant, while decrease with increasing damping coefficient. This study deepens the understanding of synchronization between coupled oscillators and may provide new design ideas for energy harvesters, soft robotics, signal detection, active motors, and self-sustained machinery.

## 1. Introduction

Self-excited oscillation is a phenomenon in which a system moves continuously in a steady-state environment, and in which the alternating force to sustain the oscillation is generally manipulated by the motion itself [1,2,3,4,5,6]. It has broad application prospects in many fields such as energy harvesting, signal monitoring, soft robotics, medical equipment, among others [7,8,9,10,11]. The periodic motion of self-excited oscillation can be maintained by periodically collecting energy from the external steady environment [12,13,14]. Because the system only needs steady external stimulation, the design of system motion control and a complex control system is easier to realize, which reduces the complexity of self-oscillating systems to a certain extent and has the advantages of portability [15,16,17,18].

In recent years, many researchers have proposed various self-oscillating systems based on active materials [17,18,19,20,21,22,23,24,25]. The stimuli-responsive materials of self-oscillating systems include hydrogels [26,27,28,29], ionic gels [30,31], and liquid crystal elastomers (LCEs) [32,33,34,35], among others. Among stimuli-responsive materials, LCE has the advantages of fast response, recoverable deformation, and low noise [36,37]. LCE is an intelligent material synthesized by liquid crystal molecules and polymer networks. Under external stimuli such as light, electricity, heat, and magnetic field, liquid crystal monomer molecules change their arrangement, resulting in macro deformation [38]. For many self-excited oscillations based on LCE, researchers have done a lot of related experimental and theoretical work [39,40,41,42,43,44]. 

Generally, a self-oscillating system requires a certain mechanism to continuously harvest energy from the external ambient to compensate for the energy dissipation of system damping [26,27,28,29,30,31,32,33,34,35]. For different stimuli-responsive materials and structures, different feedback mechanisms are proposed to realize energy compensation [45,46,47,48]. Due to a self-shading mechanism, the light-fueled self-excited oscillator based on the LCE actuator exhibits bending, twisting and contraction expansion vibration modes [49]. Based on coupling among deformation, movement and evaporation, the volatile droplets on a soft substrate can evolve into self-excited oscillation [47]. The coupling between the large deformation and reaction-diffusion of a gel layer under autocatalytic reaction can lead to self-sustained swelling and shrinking of the gel layer [50]. 

Recently, the synchronization between multiple coupled self-excited oscillators has attracted much attention. Synchronization is one of the most basic phenomena in nature, which exists all around us and has attracted the extensive attention of many researchers [51,52,53,54]. The first exploration of synchronization originated from Huygens’s clock experiment [55], which observed that two identical clocks oscillated synchronously with two pendulums in opposite directions. Recent studies have confirmed that the synchronization between the two pendulums is caused by the coupling caused by micromechanical vibration propagating in the wooden structure connecting the clocks [56,57]. In addition, the synchronous movement of a large number of metronomes with greater degrees of freedom on a freely moving base was shown experimentally [58]. Recently, Ghislaine et al. studied the synchronous oscillation of a light-driven thin plastic driver based on an optically-responsive LCE, and found that there are two synchronous oscillation phenomena of in-phase and anti-phase in the steady illumination [59]. Their numerical simulation qualitatively explained the origin of synchronous motion and found that synchronous motion can be adjusted by the mechanical properties of coupling joints.

Self-oscillators are often used as engines to drive external working components, and synchronous oscillations occur with external passive oscillators. Synchronous behaviors often have an important impact on the function of the machine. For example, the phase difference and amplitude of the self-oscillator and passive oscillator in synchronous oscillation generally depend on system factors, and may affect the work done by the self-oscillator. In this paper, we construct a new self-oscillating system composed of a passive oscillator and an active self-oscillator powered by steady illumination, in which the self-oscillator can oscillate spontaneously under steady illumination and drive the passive oscillator to oscillate synchronously. Based on a well-established dynamic LCE model [60], the synchronization of the two coupled oscillators is theoretically explored, and the effects of spring constant, light intensity, damping coefficient and other factors on synchronization are investigated. The main object of this research was to construct a coupled self-oscillating system, investigate the principles of the synchronous self-oscillation through theoretical modeling, and guide its design in the engineering applications of energy harvesters, soft robotics, signal detection, active motors, and self-sustained machinery. 

## 2. Model and Formulation

Based on the dynamic LCE model, a theoretical model of self-oscillating system composed of a passive spring oscillator and a LCE active self-oscillator powered by steady illumination was established. The model includes the dynamics of two coupled oscillators, evolution law of *cis* number fraction of LCE fiber, and nondimensionalization of the system parameters and governing equations.

### 2.1. Dynamics of Two Coupled Oscillators

Figure 1 shows a self-oscillating system composed of a passive spring oscillator and a LCE active self-oscillator powered by steady illumination. In the reference state of the LCE fiber (Figure 1a), the original lengths of LCE fiber and spring are L, and the azobenzene molecules in LCE fiber are oriented along its axis. Figure 1b depicts the initial state of the coupled oscillator. One end of the LCE fiber is fixed at the O point, and the other end is connected with the spring through a mass block m. Meanwhile, the other mass block m is hanging at the end of spring. The masses of LCE fiber and spring are negligible. ${\dot{u}}_{1}^{0}$ represents the initial speed of the upper mass block and ${\dot{u}}_{2}^{0}$ represents the initial speed of the lower mass block. The current state of the coupled oscillator is shown in Figure 1c. The illuminated zone is represented by the shaded area. $u_{1}\left(t\right)$ is the displacement of the upper mass block oscillating between the illuminated area and the non-illuminated area (the same as the material point at the end of the LCE fiber), and $u_{2}\left(t\right)$ is the displacement of the mass block below. $F_{L}\left(t\right)$ is the spring force of the LCE fiber, $F_{s}\left(t\right)$ is the spring force of the spring, and $F_{d}\left(t\right)$ is the fluid damping force applied on the small mass during the vibration process. For simplicity, we assume that $F_{d}\left(t\right)$ is proportional to the velocity of mass block. 

Initially, the self-sustained coupled oscillators are in the reference state, and an initial velocity is applied to the two mass blocks. When the upper mass vibrates in the illumination region, some LCE liquid crystal molecules change from straight *trans* conformation to curved *cis* conformation, resulting in the contraction of LCE fibers. In the non-illumination region, the change of some liquid crystal molecules from *cis* to *trans* leads to the recovery of LCE fiber length. Through the periodic contraction and relaxation of LCE fibers, the upper mass can trigger self-excited oscillation under steady illumination. In this process, the LCE fiber drives the spring and the mass block suspended at its end to oscillate periodically. At the same time, the interaction between the two mass blocks leads to the self-excited coupling oscillation system to evolve different synchronous regimes.

To analyze the inhomogeneous deformation of LCE fiber, we established a Lagrangian coordinate system X on the reference state of LCE fiber, and the Eulerian coordinate system x in its current state. Then, the instantaneous position of a material point X can be denoted as x=x(X,t). During the movement of coupled oscillator, the governing equation for the dynamics of two mass blocks can be given by
(1)$$m{\ddot{u}}_{1}=mg+F_{L}-F_{s}\left(t\right)-c{\dot{u}}_{1},$$
(2)$$m{\ddot{u}}_{2}=mg-F_{s}-c{\dot{u}}_{2},$$
where g is the gravitational acceleration, c is the damping coefficient, ${\dot{u}}_{1}$ and ${\ddot{u}}_{1}$ indicate the velocity $\frac{du_{1}\left(t\right)}{dt}$ and acceleration $\frac{d^{2}u_{1}\left(t\right)}{dt^{2}}$ of the mass block, respectively. Similarly, ${\dot{u}}_{2}$ and ${\ddot{u}}_{2}$ indicate $\frac{du_{2}\left(t\right)}{dt}$ and $\frac{d^{2}u_{2}\left(t\right)}{dt^{2}}$. The spring force $F_{s}$ can be written as
(3)$$F_{s}=k_{2}\left[u_{2}\left(t\right)-u_{1}\left(t\right)\right],$$
where $k_{2}$ is the spring constant of the spring.

For simplicity, we assume that the force of LCE fiber is linear to the deformation gradient as in the following form [61],
(4)$$F_{L}\left(t\right)=k_{1}L\left[\lambda \left(X,t\right)-1-\varepsilon \left(X,t\right)\right],$$
where $k_{1}$ is the spring constant of LCE fiber, λ(X,t) is the deformation gradient, which is written as
(5)$$\lambda \left(X,t\right)=\frac{dx\left(X,t\right)}{dX},$$
and the light-driven contraction strain ε(X,t) is assumed to be linear to the number fraction φ(X,t) of *cis* number fraction in the LCE fiber, which can be written as
(6)$$\varepsilon \left(X,t\right)=-C_{0}\varphi \left(X,t\right),$$
where $C_{0}$ is the contraction coefficient.

To obtain the instantaneous position x of the LCE fiber, $F_{L}\left(t\right)$ should be rewritten by u(t) and ε(X,t). Considering that $F_{L}\left(t\right)$ in the LCE fiber is uniform, we integrate Equation (4) from 0 to L on both sides, and obtain
(7)$$F_{L}\left(t\right)=k_{1}\left[u_{1}\left(t\right)-\int_{0}^{L}\varepsilon \left(X,t\right)dX\right].$$

Then, from Equation (4), λ(X,t) can be expressed by $F_{L}\left(t\right)$ as:(8)$$\lambda \left(X,t\right)=\frac{F_{L}\left(t\right)}{kL}+1+\varepsilon \left(X,t\right).$$

By combining Equations (5), (7) and (8), we obtain
(9)$$dx\left(X,t\right)=\left[\frac{u_{1}\left(t\right)-\int_{0}^{L}\varepsilon \left(X,t\right)dX}{L}+1+\varepsilon \left(X,t\right)\right]dX.$$

We integrate Equation (9) from 0 to X and then obtain
(10)$$x\left(X,t\right)=\frac{X}{L}\left[u_{1}\left(t\right)-\int_{0}^{L}\varepsilon \left(X,t\right)dX\right]+\int_{0}^{X}\varepsilon \left(X,t\right)dX+X.$$

The calculated x from Equation (10) can be compared with L to determine whether the LCE fiber matter point is in the illuminated or non-illuminated region.

### 2.2. Dynamic LCE Model of the LCE Fiber

To obtain the number fraction of LCE fiber in Equation (6), the dynamic LCE model was adopted. The number fraction φ(X,t) of *cis* number fraction generally depends on the thermal excitation of *trans* to *cis*, thermally driven relaxation of *cis* to *trans,* and light-driven isomerization [62]. In this work, we used the following governing equation to describe the evolution of the number fraction of *cis* number fraction [63,64],
(11)$$\frac{\partial \varphi \left(X,t\right)}{\partial t}=\eta_{0}I_{0}\left[1-\varphi \left(X,t\right)\right]-T_{0}^{-1}\varphi \left(X,t\right),$$
where $T_{0}$ is the thermal relaxation time from the *cis* state to *trans* state, $I_{0}$ is the light intensity, and $\eta_{0}$ is a light-absorption constant. 

### 2.3. Nondimensionalization

By introducing the following dimensionless parameters: $\bar{t}=t/T_{0}$, ${\bar{F}}_{L}=F_{L}T_{0}^{2}/mL$, ${\bar{F}}_{s}=F_{s}T_{0}^{2}/mL$, ${\bar{u}}_{1}=u_{1}/L$, ${\bar{u}}_{2}=u_{2}/L$, $\bar{X}=X/L$, $\bar{x}=x/L$, $\bar{c}=cT_{0}/m$, $\bar{g}=gT_{0}^{2}/L$, ${\bar{k}}_{2}k_{2}T_{0}^{2}/m$, ${\bar{k}}_{1}=k_{1}T_{0}^{2}/m$, and $\bar{I}=T_{0}\eta_{0}I_{0}$, the governing Equations (1), (2), (10) and (11) can be rewritten in dimensionless form as
(12)$${\bar{\ddot{u}}}_{1}=\bar{g}+{\bar{k}}_{2}\left[{\bar{u}}_{2}\left(\bar{t}\right)-{\bar{u}}_{1}\left(\bar{t}\right)\right]-{\bar{k}}_{1}\left[{\bar{u}}_{1}\left(\bar{t}\right)-\int_{0}^{1}\varepsilon \left(\bar{X},\bar{t}\right)d\bar{X}\right]-\bar{c}{\bar{\dot{u}}}_{1},$$
(13)$${\bar{\ddot{u}}}_{2}=\bar{g}-{\bar{k}}_{2}\left[{\bar{u}}_{2}\left(\bar{t}\right)-{\bar{u}}_{1}\left(\bar{t}\right)\right]-\bar{c}{\bar{\dot{u}}}_{2},$$
(14)$$\bar{x}\left(\bar{X},\bar{t}\right)=\bar{X}{\bar{u}}_{1}\left(\bar{t}\right)-\bar{X}\int_{0}^{1}\varepsilon \left(\bar{X},\bar{t}\right)d\bar{X}+\int_{0}^{\bar{X}}\varepsilon \left(\bar{X},\bar{t}\right)d\bar{X}+\bar{X},$$
(15)$$\frac{\partial \varphi \left(\bar{X},\bar{t}\right)}{\partial \bar{t}}=\bar{I}-\left(1+\bar{I}\right)\varphi \left(\bar{X},\bar{t}\right),$$
where ${\bar{\dot{u}}}_{1}$ and ${\tilde{\ddot{u}}}_{1}$ indicate the velocity $\frac{d{\bar{u}}_{1}\left(\bar{t}\right)}{d\bar{t}}$ and acceleration $\frac{d^{2}{\bar{u}}_{1}\left(\bar{t}\right)}{d{\bar{t}}^{2}}$ of the mass, respectively. Similarly, ${\bar{\dot{u}}}_{2}$ and ${\bar{\ddot{u}}}_{2}$ indicate $\frac{d{\bar{u}}_{2}\left(\bar{t}\right)}{d\bar{t}}$ and $\frac{d^{2}{\bar{u}}_{2}\left(\bar{t}\right)}{d{\bar{t}}^{2}}$, respectively. 

Equations (12) and (13) are ordinary differential equations with variable coefficients, and no analytical solution can be obtained. By following previous work [65], we used the classical fourth-order Runge-Kutta method to solve Equations (12)–(15) in MATLAB software, and obtained the final steady-state response of the LCE self-excited coupled oscillation system, i.e., the relationship between displacement and velocity with time histories. 

## 3. Three Synchronization Regimes and Their Mechanisms

Based on the above governing equations, we numerically studied the synchronous motion of two coupled oscillators. First, three typical synchronous regimes, namely static regime, in-phase regime and anti-phase regime, are presented. Then, the corresponding mechanisms of self-oscillations in in-phase regime and anti-phase regime are elucidated, respectively.

### 3.1. Three Synchronous Regimes

To investigate the synchronous motion of the two coupled oscillators, the typical geometric parameters and material properties were estimated for numerical calculations. According to the available experiments [10,66,67,68,69,70], we list typical values of geometric parameters and material properties of the coupled oscillators system in Table 1. Meanwhile, the corresponding dimensionless parameters are also listed in Table 2. In addition, we set the initial velocities of two mass blocks as ${\bar{\dot{u}}}_{1}^{0}=0.5$ and ${\bar{\dot{u}}}_{2}^{0}=-0.5$. Figure 2, Figure 3 and Figure 4 show the displacement time histories of the two mass blocks and the attraction domain for three different spring constants ${\bar{k}}_{2}$. In the computation, the other geometric and material parameters are given in Table 2. As shown in Figure 2a–c, the numerical results show that the two mass blocks always move in the same direction during the self-oscillations for ${\bar{k}}_{2}=16$. This means that the two oscillators are in-phase regime. As shown in Figure 3a–c, the numerical results show that the two mass blocks always move in opposite directions during the self-oscillations for ${\bar{k}}_{2}=4.5$, which indicates that the two oscillators are in anti-phase regime. Meanwhile, as shown in Figure 4a–c, the two oscillators finally develop into static regime for ${\bar{k}}_{2}=8$. In summary, there are three typical synchronous regimes for the self-sustained motion of the two mass blocks, namely, in-phase regime, anti-phase regime and static regime. Detailed numerical calculations show that the synchronous regime of self-oscillations is mainly determined by the spring constant. In the following, we further explore the mechanisms of the in-phase and anti-phase regimes and the effects of several key parameters on their amplitudes, periods, limit cycles and attraction domains.

### 3.2. Mechanisms of Self-Oscillation in the in-Phase Regime

As shown in Figure 5, to study the mechanism of self-excited oscillation in the in-phase regime of two oscillators under steady illumination, the relationship diagrams of some key physical quantities in the in-phase regime in Figure 2 are given. The shadow area in the figure indicates that LCE fiber is in the illuminated region. During the oscillation of the upper mass block, the *cis*-number fraction of the material point at the end of the LCE fiber increases in the illuminated region and decreases in the non-illuminated region, as shown in Figure 5a. As a result, light-triggered contraction increases in the illuminated region and decreases in the non-illuminated region, as shown in Figure 5b. Since the number fraction of LCE fibers and the photo-triggered contraction show periodic changes, the spring force magnitude of LCE fibers first increases and then decreases in the illuminated region, as shown in Figure 5c. In Figure 5d, the spring force of the LCE fiber and the displacement of the upper mass block shows a closed-loop relationship, and the area enclosed by the closed-loop represents the net work done by the force. Under the action of periodic contraction and relaxation of the LCE fiber, the spring force magnitude of the spring also shows periodic changes, as shown in Figure 5e. In Figure 5f, the spring force of the spring and the displacement of the lower mass block show a closed-loop relationship, and the area enclosed by the closed-loop represents the net work done by the force. Once the net work of the spring force is equal to the energy consumed by the damping of the lower mass, its self-excited oscillation can be maintained. Since the two mass blocks can attain self-excited oscillation respectively, and because ${\bar{k}}_{2}=16$, the coupling between the two mass blocks is strong, and the synchronous motion regime of the coupling oscillator is in-phase.

### 3.3. Mechanisms of Self-Oscillation in the Anti-Phase Regime

As shown in Figure 6, the relationship diagrams of some key physical quantities in the anti-phase regime in Figure 3 are obtained by numerical calculations. During the oscillation of upper mass block, the *cis* number fraction of material point at the end of LCE fiber increases in the illuminated region and decreases in the non-illuminated region, as shown in Figure 6a. As a result, the light-triggered contraction increases in the illuminated region and decreases in the non-illuminated region, as shown in Figure 6b. Similarly, the spring force of LCE fiber first increases and then decreases in the illuminated region, as shown in Figure 6c. In Figure 6d, the spring force of LCE fiber and the displacement of upper mass block show a closed-loop relationship, and the area enclosed by the closed loop represents the net work done by the force. Under the action of periodic contraction and relaxation of the LCE fiber, the spring force of the spring also changes periodically as shown in Figure 6e. In Figure 6f, the spring force of the spring and the displacement of the lower mass block show a closed-loop relationship, and the area enclosed by the closed-loop represents the net work done by the force. 

## 4. Parametric Study

Generally, the dynamics of two coupled oscillators depends on the dimensionless parameters of system in Table 2. In this section, we investigate the effects of several typical systematic parameters on the synchronous regime, amplitudes and limit cycles of self-oscillations of two coupled oscillators in detail. 

### 4.1. Effect of Spring Constant of Spring

Figure 7 shows the effect of spring constant ${\bar{k}}_{2}$ of the spring on the self-oscillations of two oscillators. In the computation, the other geometric and material parameters are given in Table 2. In Figure 7a,c,e, for ${\bar{k}}_{2}>9.5$, the self-oscillations of two oscillators evolve into an in-phase regime, and the amplitudes of self-oscillations monotonously increase with the increase of spring constant ${\bar{k}}_{2}$. For $7\leq {\bar{k}}_{2}\leq 9.5$, the amplitudes of self-oscillations gradually decay with the decrease of spring constant ${\bar{k}}_{2}$, and finally evolve into a static regime, as shown in Figure 7c,e. In Figure 7b,d,f, for ${\bar{k}}_{2}<7$, the self-oscillations of two oscillators evolve into an anti-phase regime, and the amplitudes of self-oscillations gradually decrease with increases of the spring constant ${\bar{k}}_{2}$. For $7\leq {\bar{k}}_{2}\leq 9.5$, the amplitudes of self-oscillations gradually decay with the increase of spring constant ${\bar{k}}_{2}$, and the system finally evolves into static regime, as shown in Figure 7d,f. The reason for this phenomenon is that in the in-phase regime, with the increase of spring constant, the spring forces of spring and LCE fiber gradually increase, and their corresponding net works increases. Therefore, the self-oscillations of two coupled oscillators can be maintained. In the anti-phase regime, as the spring force increases, the tension of LCE fiber gradually decreases and its corresponding net work decreases. Therefore, the synchronous motion of two coupled oscillators cannot be maintained. 

In Figure 7e,f, it is noted that with the decrease of spring constant ${\bar{k}}_{2}$ from 7 to 9.5, the position coordinates of the lower mass block at rest will gradually increase. This is because that with the decreasing spring constant ${\bar{k}}_{2}$, the spring length in the static state is greater, and the equilibrium position of lower mass block changes. Meanwhile, the numerical results show that only the spring constant ${\bar{k}}_{2}$ determines the synchronous regime of self-oscillations of two oscillators. The two oscillators oscillate in an in-phase regime for ${\bar{k}}_{2}<7$, and oscillate in an anti-phase regime for ${\bar{k}}_{2}>9.5$. Meanwhile, the system is in the static regime for $7\leq {\bar{k}}_{2}\leq 9.5$. In the following, we describe the influences of other parameters on the self-oscillation of two oscillators in the in-phase regime of ${\bar{k}}_{2}=16$ and in the anti-phase regime of ${\bar{k}}_{2}=4.5$.

### 4.2. Effect of Damping Coefficient

Figure 8 shows the effect of the damping coefficient $\bar{c}$ on the self-oscillations of two oscillators for the in-phase regime of ${\bar{k}}_{2}=16$ and the anti-phase regime of ${\bar{k}}_{2}=4.5$. In the computation, the other geometric and material parameters are given in Table 2. As shown in Figure 8a,b, the damping coefficient does not affect the synchronization regime of two oscillators, which is mainly determined by the spring constant ${\bar{k}}_{2}$, as discussed in Section 4.1. In the in-phase regime, the amplitudes of self-oscillations of the two oscillators decrease with the increase of damping coefficient, and the limit cycles finally evolve into the static regime, as shown in Figure 8c,e. The self-oscillations of the two oscillators can be triggered for $\bar{c}<0.29$, while the oscillation is eventually suppressed for $\bar{c}\geq 0.29$. In the anti-phase regime, for $\bar{c}<0.3$, the two coupled oscillators can self-oscillate, and the amplitudes of self-oscillations increase with the decrease of damping coefficient. For $\bar{c}\geq 0.3$, the limit cycles evolve into the static regime, as shown in Figure 8d,f. 

The influence of damping coefficient on the in-phase and anti-phase regimes is the same. With increase of the damping coefficient, the amplitudes of self-oscillations of the two mass blocks gradually decrease. The reason for this phenomenon is that due to the increase in damping coefficient, the light energy input by the external environment does not compensate for damping dissipation, and the self-oscillation of mass block is gradually suppressed. Therefore, the system cannot maintain the self-sustained synchronous motion of two coupled oscillators.

### 4.3. Effect of Contraction Coefficient

Figure 9 shows the effect of the contraction coefficient $C_{0}$ on the self-oscillations of two oscillators for the in-phase regime of ${\bar{k}}_{2}=16$ and the anti-phase regime of ${\bar{k}}_{2}=4.5$. In the computation, the other geometric and material parameters are given in Table 2. It can be seen from Figure 9a,b that the contraction coefficient does not affect the synchronization regime of the two oscillators. In the in-phase regime, the amplitudes of self-oscillations of two oscillators gradually decrease with the decrease of contraction coefficient, and the limit cycles finally evolve into the static regime, as shown in Figure 9c,e. The self-oscillation of the two oscillators can be triggered for $C_{0}>0.37$, while the oscillation regime is eventually suppressed for $C_{0}<0.37$. In the anti-phase regime, it can be seen that for $C_{0}>0.3$, with the increase of contraction coefficient the amplitude of self-oscillation increases gradually. However, for $C_{0}\leq 0.3$, the limit cycles evolve into the static regime, which is denoted by the point in Figure 9d,f. 

Similarly, the influences of the contraction coefficient on the in-phase and anti-phase regimes are the same. With the decrease of contraction coefficient, the amplitudes of self-oscillations of two mass blocks gradually decrease, and the self-excited oscillation is gradually suppressed. This phenomenon can be understood being due to the reduction of contraction, the deformation of fiber, and the force of fiber, the net work done by the system in the coupling process of deformation and vibration decrease, which cannot compensate for the dissipation of system damping. Therefore, the self-excited oscillation of two mass blocks cannot be maintained.

### 4.4. Effect of Light Intensity

Figure 10 shows the effect of light intensity $\bar{I}$ on the self-oscillations of two oscillators for the in-phase regime of ${\bar{k}}_{2}=16$ and the anti-phase regime of ${\bar{k}}_{2}=4.5$. In the computation, the other geometric and material parameters are given in Table 2. The numerical results show that the light intensity does not affect the synchronization regime of the two oscillators, as shown in Figure 10a,b. In the in-phase regime, the amplitudes of self-oscillations gradually decrease with the decrease of light intensity, and the limit cycles finally evolve into the static regime, as shown in Figure 10c,e. This shows that the self-oscillation of two oscillators can be triggered for $\bar{I}>0.9$, while the oscillation regime is eventually suppressed for $\bar{I}\leq 0.9$. In the anti-phase regime, it can be seen that for $\bar{I}>0.9$, with the increase of light intensity, the amplitudes of self-oscillations increase gradually. However, for $\bar{I}\leq 0.9$, the limit cycles evolve into the static regime, as shown in Figure 10d,f. 

In addition, the influences of light intensity on the in-phase and anti-phase regimes are the same. With the decrease of light intensity, the amplitudes of self-excited oscillations of the two mass blocks gradually decrease, and the self-excited oscillation is finally suppressed. This phenomenon can be explained as follows. With the gradual decrease of light intensity, the light-driven contraction of LCE fiber decreases gradually, resulting in the gradual decrease of spring force and the net work caused by light-driven contraction, and the net work done by spring force also decreases gradually. Therefore, the dissipation of system damping cannot be compensated, and the self-excited oscillation of the two mass blocks cannot be maintained.

### 4.5. Effect of Initial Velocity

Figure 11 shows the effect of initial velocity ${\bar{\dot{u}}}_{1}^{0}$ on the self-oscillations of two oscillators for the in-phase regime of ${\bar{k}}_{2}=16$ and the anti-phase regime of ${\bar{k}}_{2}=4.5$. In the computation, the other geometric and material parameters are given in Table 2. The numerical results show that the initial velocity ${\bar{\dot{u}}}_{1}^{0}$ does not affect the synchronization regime of the two oscillators, as shown in Figure 11a,b. In the in-phase regime, the initial velocity ${\bar{\dot{u}}}_{1}^{0}$ does not affect the limit cycle and amplitude of self-oscillation of two oscillators, as shown in Figure 11c,e. In the anti-phase regime, it can be seen that the initial velocity ${\bar{\dot{u}}}_{1}^{0}$ also does not affect the limit cycle and amplitude of self-oscillation of two oscillators, as shown in Figure 11d,f.

In addition, Figure 12 shows the effect of initial velocity ${\bar{\dot{u}}}_{2}^{0}$ on the self-oscillations of two oscillators for the in-phase regime of ${\bar{k}}_{2}=16$ and the anti-phase regime of ${\bar{k}}_{2}=4.5$. In the computation, the other geometric and material parameters are given in Table 2. The numerical results show that the initial velocity ${\bar{\dot{u}}}_{2}^{0}$ does not affect the synchronization regime of two oscillators, as shown in Figure 12a,b. It is seen that the initial velocity ${\bar{\dot{u}}}_{2}^{0}$ does not affect the limit cycles and amplitude of the self-oscillations of two oscillators in synchronous regime, as shown in Figure 12c–f. This is because the initial condition does not change the energy input of the system and the net work done by fiber in the coupling process of fiber contraction and oscillation of the two mass blocks. In conclusion, the initial velocity does not affect the synchronous regime and inherent characteristics of self-oscillation of the two oscillators. The effects of initial velocity on self-oscillations of two coupled oscillators based on LCE fiber are the same as that in other self-oscillating systems [57].

## 5. Conclusions

Self-oscillators capable of harvesting energy from a steady environment can be adopted as an engine to drive additional working equipment, and the synchronous behavior of self-oscillators and passive oscillators may have an important impact on their functions. In this research, based on optically-responsive LCE fibers, we constructed a self-oscillating system composed of a passive oscillator and an active self-oscillator powered by steady illumination, and theoretically investigated the synchronization of two coupled oscillators. The governing equations of dynamics of the two coupled oscillators under steady illumination were derived and numerical calculations were performed in Matlab software. The results show that the movement of two oscillators always develops into in-phase regime, anti-phase regime, or static regime. In the process of self-oscillation, the *cis* number fraction and the light-driven contraction in the LCE fiber, and the tensions of the LCE fiber and spring, change periodically and continuously. Self-oscillation is maintained by energy input from the environment to compensate for damping dissipation. The self-oscillations of the two oscillators can be triggered by adjusting the spring constant, damping coefficient, light intensity, and contraction coefficient, and the synchronous regime is mainly determined by the spring constant. The amplitudes of self-oscillations of the two oscillators increase with the increase of contraction coefficient, light intensity, and the spring constant of spring, and decrease with the increase of damping coefficient. These results deepen the understanding of the synchronization behaviors of coupled oscillators and provide new design ideas for energy harvesters, soft robotics, signal detection, active motors, and self-sustained machinery.

## Figures and Tables

**Figure 1 polymers-14-03058-f001:**
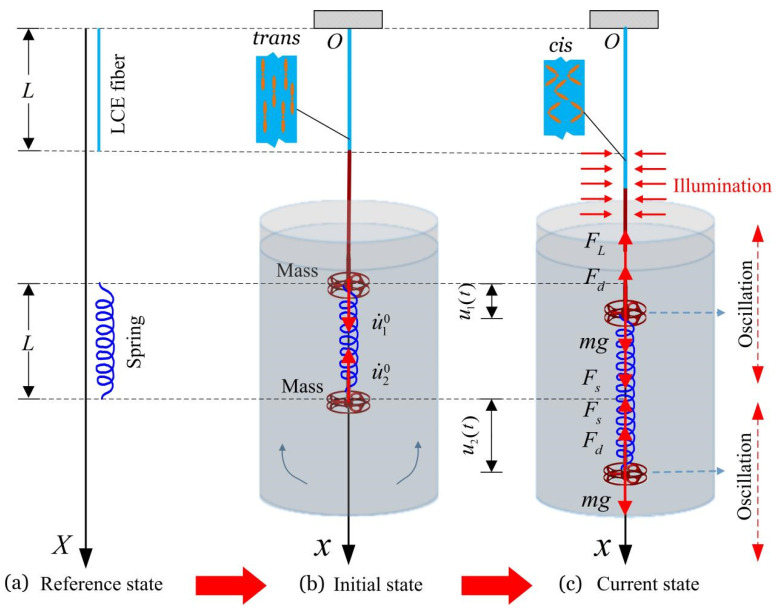
Schematic of a self-oscillating system composed of a passive spring oscillator and an LCE active self-oscillator powered by steady illumination. Both the LCE fiber and spring are connected to each mass block. The two mass blocks are placed into a fluid, and the damping coefficient can be easily tuned by controlling the viscosity of the fluid. Under steady illumination, the two coupled oscillators may vibrate synchronously.

**Figure 2 polymers-14-03058-f002:**
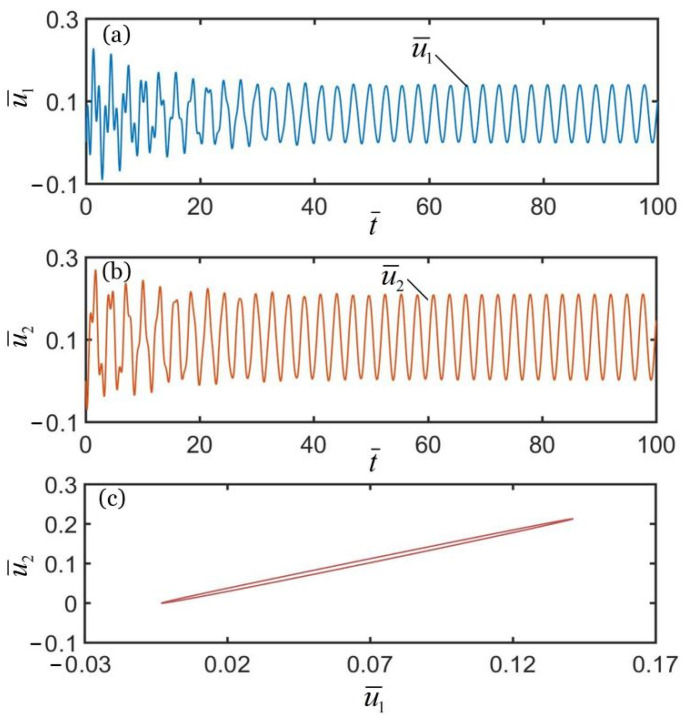
Time histories and domain of attraction of in-phase synchronous regime of light-powered LCE self-excited coupled oscillators for ${\bar{k}}_{2}=16$. (**a**) ${\bar{u}}_{1}$ vs. $\bar{t}$; (**b**) ${\bar{u}}_{2}$ vs. $\bar{t}$; (**c**) domain of attraction. The two couple oscillators are in in-phase regime.

**Figure 3 polymers-14-03058-f003:**
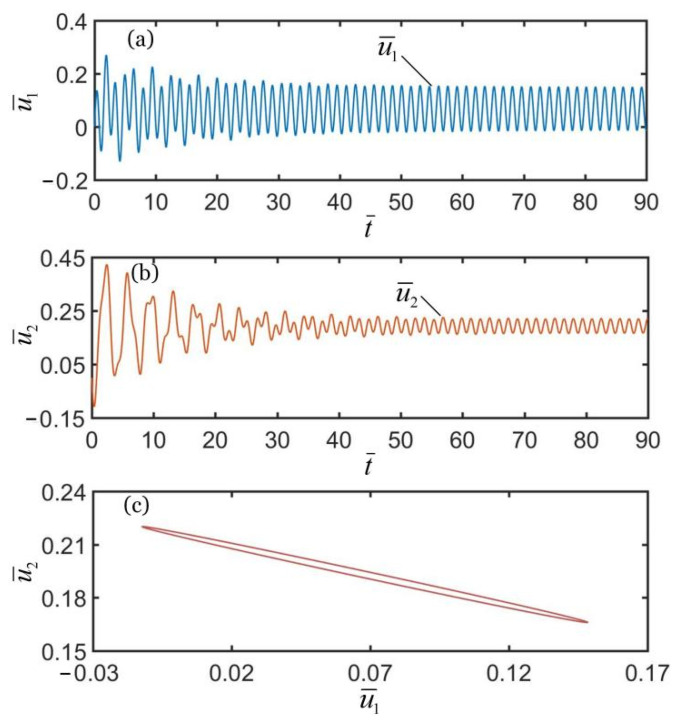
Time histories and domain of attraction of anti-phase synchronous regime of light-powered LCE self-excited coupled oscillators for ${\bar{k}}_{2}=4.5$. (**a**) ${\bar{u}}_{1}$ vs. $\bar{t}$; (**b**) ${\bar{u}}_{2}$ vs. $\bar{t}$; (**c**) domain of attraction. The two couple oscillators are in anti-phase regime.

**Figure 4 polymers-14-03058-f004:**
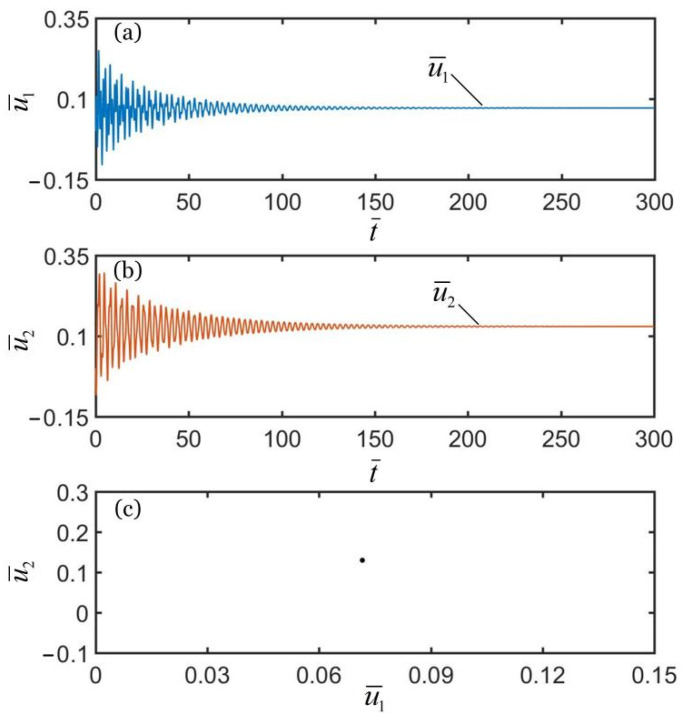
Time histories and domain of attraction of in-phase synchronous regime of light-powered LCE self-excited coupled oscillators for ${\bar{k}}_{2}=8$. (**a**) ${\bar{u}}_{1}$ vs. $\bar{t}$; (**b**) ${\bar{u}}_{2}$ vs. $\bar{t}$; (**c**) domain of attraction. The two couple oscillators are in static regime.

**Figure 5 polymers-14-03058-f005:**
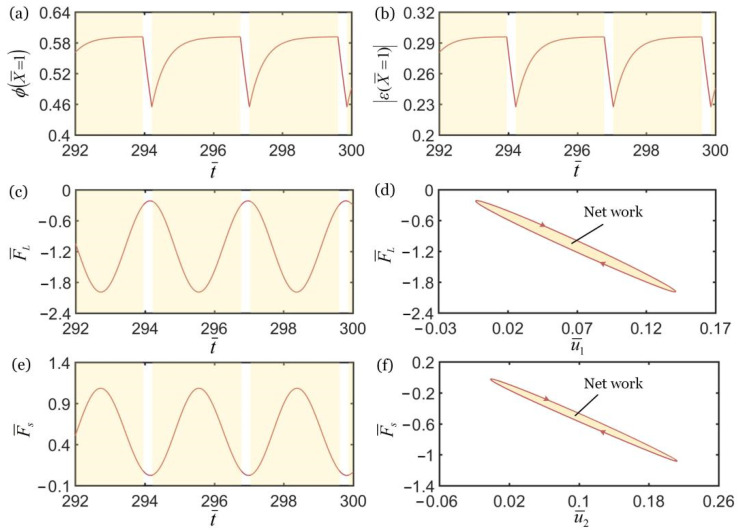
Mechanism of self-excited oscillation in in-phase regime of ${\bar{k}}_{2}=16$ in Figure 2. (**a**) $\varphi \left(\bar{X}=1\right)$ vs. $\bar{t}$. (**b**) $\left|\varepsilon \left(\bar{X}=1\right)\right|$ vs. $\bar{t}$. (**c**) ${\bar{F}}_{L}$ vs. $\bar{t}$. (**d**) ${\bar{F}}_{L}$ vs. ${\bar{u}}_{1}$. (**e**) ${\bar{F}}_{s}$ vs. $\bar{t}$. (**f**) ${\bar{F}}_{s}$ vs. ${\bar{u}}_{2}$. In Figure 3d,f, the area enclosed by the closed loop represents the net work done by the tensions of the LCE fiber and spring, which compensates for the damping dissipation to maintain the oscillations of two coupled oscillators.

**Figure 6 polymers-14-03058-f006:**
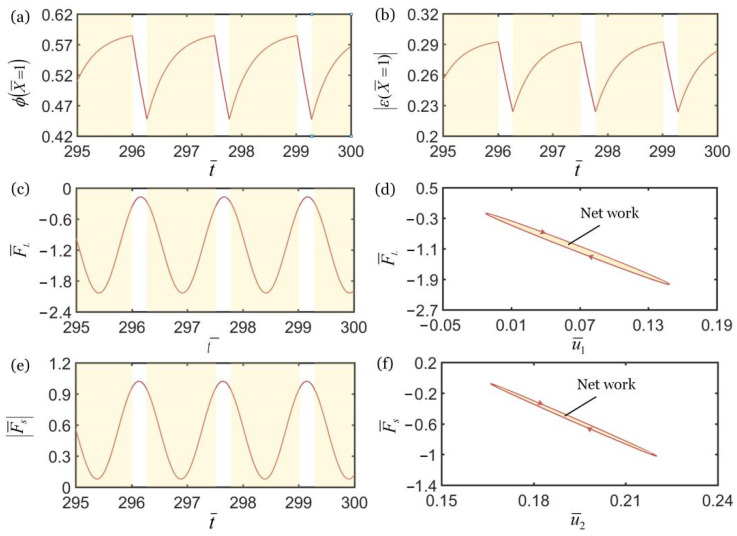
Mechanism of self-excited oscillation in anti-phase regime of ${\bar{k}}_{2}=4.5$ in Figure 3. (**a**) $\varphi \left(\bar{X}=1\right)$ vs. $\bar{t}$. (**b**) $\left|\varepsilon \left(\bar{X}=1\right)\right|$ vs. $\bar{t}$. (**c**) ${\bar{F}}_{L}$ vs. $\bar{t}$. (**d**) ${\bar{F}}_{L}$ vs. ${\bar{u}}_{1}$. (**e**) ${\bar{F}}_{s}$ vs. $\bar{t}$. (**f**) ${\bar{F}}_{s}$ vs. ${\bar{u}}_{2}$. In Figure 4d,f, the area enclosed by the closed loop represents the net work done by the tensions of LCE fiber and spring, which compensates for the damping dissipation to maintain the oscillations of two coupled oscillators.

**Figure 7 polymers-14-03058-f007:**
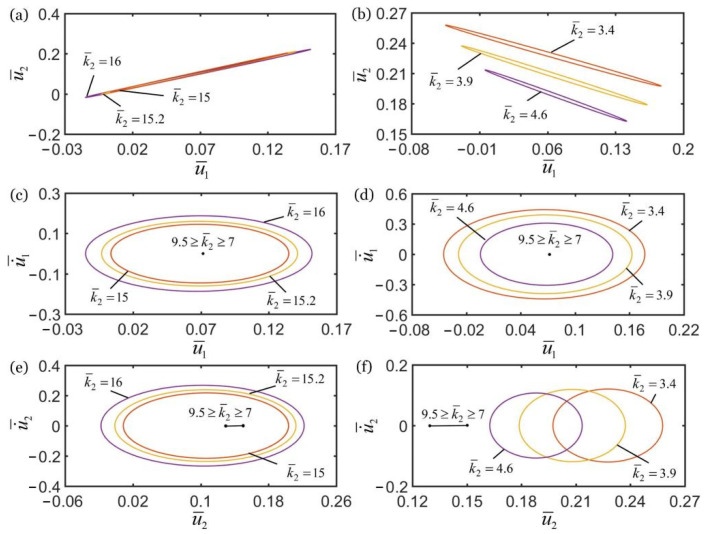
The effect of spring constant ${\bar{k}}_{2}$ on the self-oscillations of two coupled oscillators. In the computation, the other geometric and material parameters are given in Table 2. (**a**) Domain of attraction, and (**c**,**e**) limit cycles for in-phase regimes of ${\bar{k}}_{2}>9.5$. (**b**) Domain of attraction, and (**d**,**f**) limit cycles for anti-phase regimes of ${\bar{k}}_{2}<7$. The synchronous regime of self-oscillations of two oscillators is mainly determined by the spring constant.

**Figure 8 polymers-14-03058-f008:**
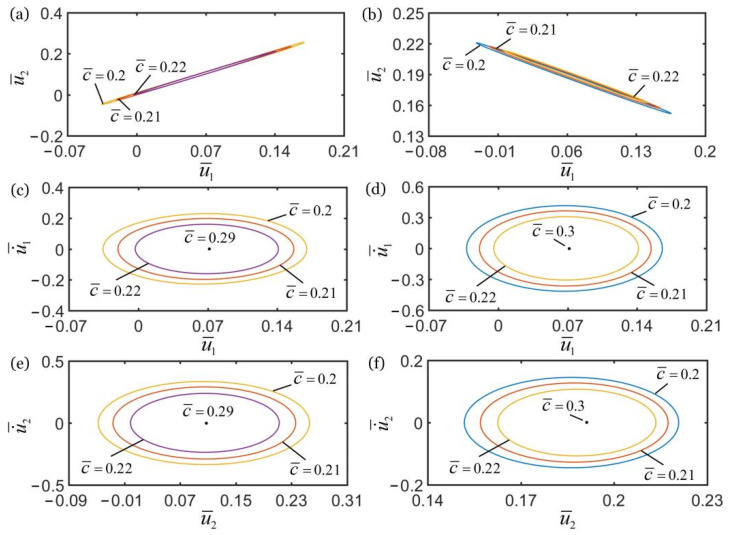
The effect of damping coefficient $\bar{c}$ on self-oscillations of the two coupled oscillators. (**a**) Domain of attraction, and (**c**,**e**) limit cycles for in-phase regime of ${\bar{k}}_{2}=16$. (**b**) Domain of attraction, and (**d**,**f**) limit cycles for anti-phase regime of ${\bar{k}}_{2}=4.5$. In the computation, the other geometric and material parameters are given in Table 2. With the increase of $\bar{c}$, the amplitudes of self-oscillations of two coupled oscillators decrease.

**Figure 9 polymers-14-03058-f009:**
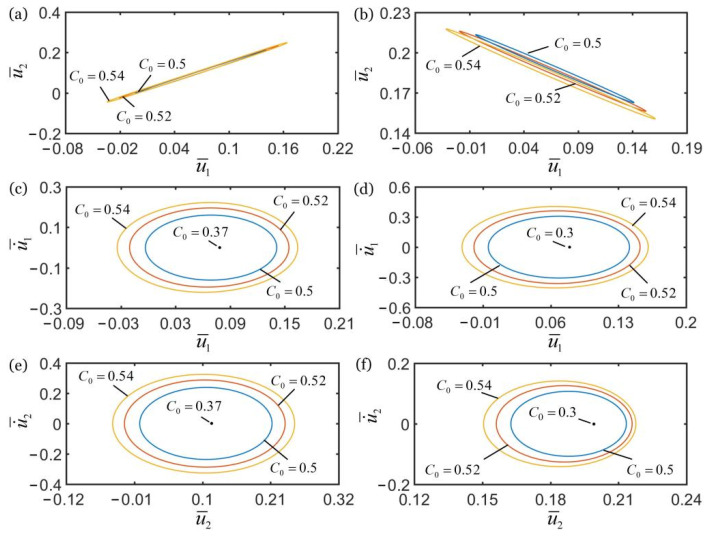
The effect of contraction coefficient $C_{0}$ on self-oscillations of the two coupled oscillators. (**a**) Domain of attraction, and (**c**,**e**) limit cycles for in-phase regime of ${\bar{k}}_{2}=16$. (**b**) Domain of attraction, and (**d**,**f**) limit cycles for anti-phase regime of ${\bar{k}}_{2}=4.5$. In the computation, the other geometric and material parameters are given in Table 2. With the increase of $C_{0}$, the amplitudes of self-oscillations of two coupled oscillators increase.

**Figure 10 polymers-14-03058-f010:**
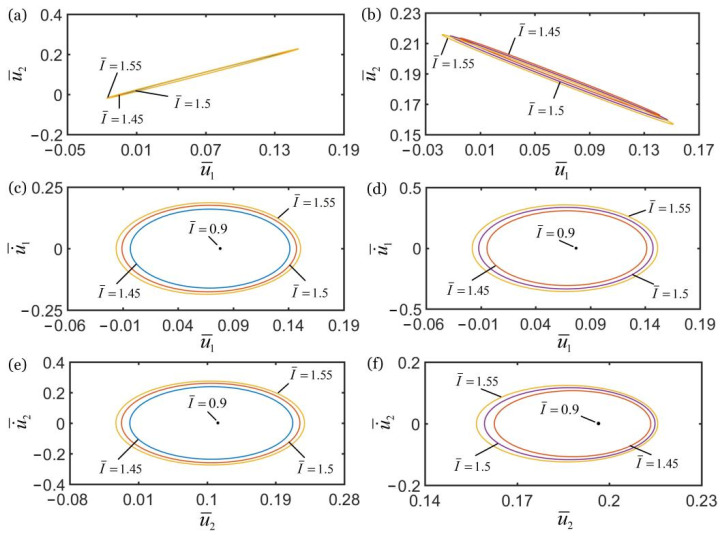
The effect of light intensity $\bar{I}$ on self-oscillations of the two coupled oscillators. (**a**) Domain of attraction, and (**c**,**e**) limit cycles for in-phase regime of ${\bar{k}}_{2}=16$. (**b**) Domain of attraction, and (**d**,**f**) limit cycles for anti-phase regime of ${\bar{k}}_{2}=4.5$. In the computation, the other geometric and material parameters are given in Table 2. With the increase of $\bar{I}$, the amplitudes of self-oscillations of two coupled oscillators increase.

**Figure 11 polymers-14-03058-f011:**
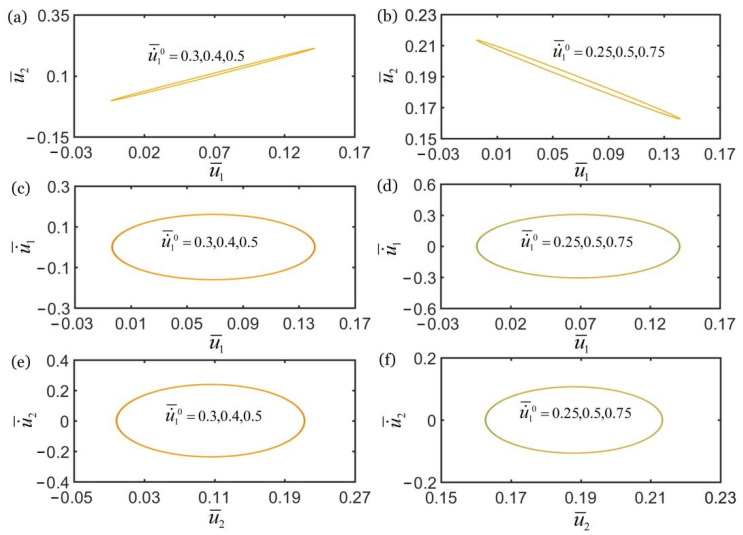
The effect of initial velocity ${\bar{\dot{u}}}_{1}^{0}$ on self-oscillations of the two coupled oscillators. (**a**) Domain of attraction, and (**c**,**e**) limit cycles for in-phase regime of ${\bar{k}}_{2}=16$. (**b**) Domain of attraction, and (**d**,**f**) limit cycles for anti-phase regime of ${\bar{k}}_{2}=4.5$. In the computation, the other geometric and material parameters are given in Table 2. The initial velocity ${\bar{\dot{u}}}_{1}^{0}$ does not affect the self-oscillations of two oscillators.

**Figure 12 polymers-14-03058-f012:**
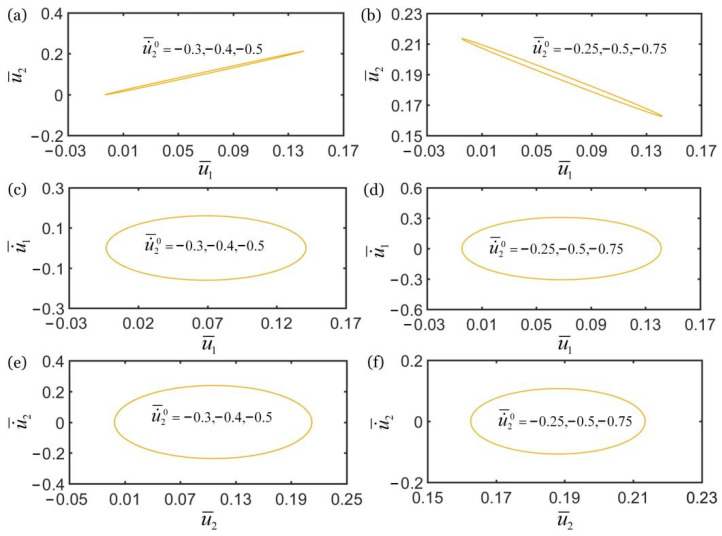
The effect of initial velocity ${\bar{\dot{u}}}_{2}^{0}$ on self-oscillations of the two coupled oscillators. (**a**) Domain of attraction, and (**c**,**e**) limit cycles for in-phase regime of ${\bar{k}}_{2}=16$. (**b**) Domain of attraction, and (**d**,**f**) limit cycles for anti-phase regime of ${\bar{k}}_{2}=4.5$. In the computation, the other geometric and material parameters are given in Table 2. The initial velocity ${\bar{\dot{u}}}_{2}^{0}$ does not affect the self-oscillations of two oscillators.

**Table 1 polymers-14-03058-t001:** Material properties and geometric parameters.

Parameter	Definition	Value	Units
m	Mass of each mass block	0.01	kg
L	Original length of LCE fiber and spring	0.18	m
g	Gravitational acceleration	10	${m/s}^{2}$
$k_{1}$	Spring constant of LCE fiber	9.5	N/m
$k_{2}$	Spring constant of spring	4.4~16	N/m
$T_{0}$	cis-to-trans thermal relaxation time of LCE fiber	0.1	s
c	Damping coefficient	0.022	kg/s
$C_{0}$	Contraction coefficient of LCE fiber	0.5	/

**Table 2 polymers-14-03058-t002:** Dimensionless parameters.

Parameter	${\bar{k}}_{1}$	$\bar{g}$	$\bar{c}$	$\bar{I}$	$C_{0}$	${\bar{\dot{u}}}_{1}^{0}$	${\bar{\dot{u}}}_{2}^{0}$	${\bar{k}}_{2}$
Value	11.2	0.55	0.22	1.45	0.5	0.5	−0.5	4.4~16

## Data Availability

The data that support the findings of this study are available upon reasonable request from the authors.

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
