# Peer review of "Synchronization of a Passive Oscillator and a Liquid Crystal Elastomer Self-Oscillator Powered by Steady Illumination"

_polymers, 2022, doi:10.3390/polym14153058_

Round 1
Reviewer 1 Report
This manuscript presented a study about the synchronization of a passive oscillator and a liquid crystal elastomer self-oscillator powered by steady illumination. The work has some potential. However, several points listed below need to be improved.
Abstract: I suggest add numerical results to the abstract.
Lines 76-84 - Introduction: I suggest remove the description of each manuscript section from the introduction.
Introduction: what is the novelty of this work? I suggest clearer the novelty of this study in the final paragraph of the introduction section.
Section 3: Is the parameters presented in Table 1 related to which material? The authors must better describe the parameters presented in Table 1.
Table 2: How the dimensionless parameters were chosen?
Figure 3 and Figure 4: the results presented in both figures must be discussed in the manuscript.
Reviewer 2 Report
The manuscript entitled <Synchronization of a passive oscillator and a liquid crystal elastomer self-oscillator powered by steady illumination> is a valid research work with appropriate level of novelty and originality. The topic of the manuscript fits well the scope of the journal. The introduction section clearly describes the state-of-the-art in the specific field of research. The technical quality of the manuscript is high. The main results are described and treated correctly and they go far beyond the state-of-the-art. The manuscript should be of reasonable interest for the readers dealing with smart responsive systems. The results presented in this work will have an impact on both the basic science and applied research. The language of the manuscript is very clear. I do not have any specific comments. Overall, with high pleasure I can recommend this manuscript for acceptance at Polymers-MDPI journal as it is.
Author Response
We appreciate the reviewer’s comments. We are also grateful that the reviewer likes our work in general.
Round 2
Reviewer 1 Report
After corrections the manuscript reads well. I suggest publication in its current form.